# Combining Unimanual and Bimanual Therapies for Children with Hemiparesis: Is There an Optimal Delivery Schedule?

**DOI:** 10.3390/bs13060490

**Published:** 2023-06-09

**Authors:** Ka Lai K. Au, Julie L. Knitter, Susan Morrow-McGinty, Talita C. Campos, Jason B. Carmel, Kathleen M. Friel

**Affiliations:** 1Blythedale Children’s Hospital, Valhalla, NY 10595, USA; 2School of Nursing, Columbia University Irving Medical Center, New York, NY 10032, USA; 3Weinberg Family Cerebral Palsy Center, Department of Neurology, Columbia University Irving Medical Center, New York, NY 10032, USA; 4Burke Neurological Institute, White Plains, NY 10605, USA; 5Brain Mind Research Institute, Weill Cornell Medical College, New York, NY 10021, USA

**Keywords:** hemiplegia, occupational therapy, cerebral palsy, constraint

## Abstract

Constraint-induced movement therapy (CIMT) and bimanual therapy (BT) are among the most effective hand therapies for children with unilateral cerebral palsy (uCP). Since they train different aspects of hand use, they likely have synergistic effects. The aim of this study was to examine the efficacy of different combinations of mCIMT and BT in an intensive occupational therapy program for children with uCP. Children (*n* = 35) participated in intensive modified CIMT (mCIMT) and BT, 6 weeks, 5 days/week, 6 h/day. During the first 2 weeks, children wore a mitt over the less-affected hand and engaged in functional and play activities with the affected hand. Starting in week 3, bimanual play and functional activities were added progressively, 1 hour/week. This intervention was compared to two different schedules of block interventions: (1) 3 weeks of mCIMT followed by 3 weeks of BT, and (2) 3 weeks of BT followed by 3 weeks of mCIMT. Hand function was tested before, after, and two months after therapy with the Assisting Hand Assessment (AHA), Pediatric Evaluation of Disability Inventory (PEDI), and Canadian Occupational Performance Measure (COPM). All three groups of children improved in functional independence (PEDI; *p* < 0.031), goal performance (COPM Performance; *p* < 0.0001) and satisfaction (COPM Satisfaction; *p* < 0.0001), which persisted two months post-intervention. All groups showed similar amounts of improvement, indicating that the delivery schedule for mCIMT and BT does not significantly impact the outcomes.

## 1. Introduction

Intensive hand therapy is among the most effective, evidence-based therapies for children with hemiplegia [1]. A key question in hemiplegia therapy is whether the affected hand should be trained alone or in tandem with the other hand. In constraint-induced movement therapy (CIMT), a participant’s less-affected upper extremity is restricted with a sling, cast, or mitt, while the participant actively uses the affected arm and hand in skill-based therapeutic activities [2]. Bimanual training (BT), in contrast, engages both hands in therapeutic movement [3]. These two interventions have shown equivalence in most studies and Cochrane reviews [4,5,6,7,8,9,10], though some studies show that BT is more effective in improving functional use of the affected hand [9,11]. Since most functional activities of daily living require bimanual coordination, BT is thought to be most effective at improving performance of these activities [12]. Alternatively, there is evidence that CIMT may be more effective in home- and school-based environments [13], isolated hand movements [14], and may produce stronger improvements in head and reach control [15] than BT.

Since CIMT and BT target different aspects of hand use, they likely have synergistic effects on hand function [16]. By requiring children to use the more-affected arm, CIMT is especially useful for overcoming “developmental disuse” [2]. The sensorimotor experience of CIMT improves function and may “prime” the affected hand by increasing a child’s awareness and engagement of the affected hand. This may then make subsequent bimanual therapy more effective. A combined approach of CIMT followed by bimanual therapy has been found to improve outcomes [17] beyond CIMT alone [18]. 

While most therapies have directly compared CIMT and BT, either by comparing one therapy against the other or administering both therapies in a block design, other combinations have not been well-studied. We developed a six-week occupational therapy program that combines CIMT and BT over the course of treatment. The goal was to engage the affected arm and hand using modified constraint-induced movement therapy (mCIMT), then to focus on functional activities that require both hands. We rationalized that an optimal therapy would first increase awareness and engagement of the impaired arm and hand, then train that hand in functional bimanual tasks. We compared this approach to two groups who received blocks of interventions. One of these groups completed three weeks of mCIMT, followed by three weeks of BT, following the same rationale of first focusing on improving the skill of the impaired upper limb and then introducing bimanual training. For a comparison, one other group completed three weeks of BT, followed by three weeks of mCIMT. This group was included to control for the order of the types of training. The goal of this study was to determine if efficacy is impacted by the schedule of delivery of mCIMT versus BT. We hypothesized that children who received mCIMT before BT would show a greater improvement in hand function at the end of the intervention, as focusing on strengthening the more-affected hand first would optimize the efficacy of the subsequent BT.

## 2. Materials and Methods

### 2.1. Participants

Thirty-five children participated in the study during the summers of 2011–2018. Five to eleven children participated each summer. Demographics and clinical characteristics of participants are shown in Table 1. Children were recruited from the outpatient service at Blythedale Children’s Hospital and through community outreach. 

Inclusion criteria: (1) unilateral brain injury resulting in impairment of one side of the body, (2) ability to move all joints of affected upper extremity, (3) age 4–12 years, and (4) ability to comply with study protocol. Exclusion criteria: health problems or uncorrected vision that would interfere with study participation. The study was approved by the Institutional Review Board of Blythedale Children’s Hospital. 

The interventions were offered as part of a clinical program in the Department of Occupational Therapy. IRB approval was obtained to analyze data from the intervention for the group that received the blended intervention. Children in the crossover groups were prospectively enrolled and provided written assent. Their caregiver provided written informed consent. The study was registered on clinicaltrials.gov (NCT02840643) before the first child was randomized to one of the crossover groups.

### 2.2. Interventions

All intervention groups used combinations of modified constraint-induced movement therapy (mCIMT) and bimanual therapy (BT). Therapy was conducted in a large room, such that all children and occupational therapists had the opportunity to interact throughout the program. The program was coordinated by an experienced occupational therapist, who was in the therapy room during the duration of the program. Each intervention took place for 6 h per day, 5 days per week, for 6 weeks (180 h).

Three different interventions were tested:3 weeks of mCIMT followed by 3 weeks of BT (group CB);3 weeks of BT followed by 3 weeks of mCIMT (group BC);2 weeks of mCIMT followed by stepwise incorporation of BT, increasing the amount of BT by 1 h per day for each of the next 4 weeks (group Step).

Materials: Both mCIMT and BT used toys, board games, art supplies, craft supplies, and sports equipment selected and structured by occupational therapists. Children brought items to the intervention for practicing caregiver/child-identified self-care and functional goals, such as a shirt with buttons or shoes with laces.

Participants engaged in age-appropriate training 6 h/day for 30 days (180 h).

Providers and Location: Therapy was provided in one room at a pediatric rehabilitation center. Therapy was provided by occupational therapists. The ratio of therapists to children was approximately 1:4. For 60 min daily, each child received 1:1 training with an OT. In addition to the OTs, there were 1–2 volunteers and/or OT interns always present in the therapy room. 

Duration of therapy and therapy regimen: Therapy was provided 6 h/day for six weeks (30 days, 180 h total). The CB group received three weeks of mCIMT, followed by three weeks of BT. The BC group received three weeks of BT, followed by three weeks of mCIMT. 

The Step group had a stepwise integration of BT after starting the intervention with mCIMT. During the first two weeks of the intervention, children received mCIMT. During the third week of the intervention, children received mCIMT for the first five hours of each day, then received one hour of BT. In each subsequent week of intervention, the duration of mCIMT was reduced by one hour per day, while the duration of BT was increased by one hour per day. Thus, in week four of the intervention, children began each day with four hours of mCIMT, followed by two hours of BT. In week five of the intervention, children began each day with three hours of mCIMT, followed by three hours of BT. In the sixth and final week of the intervention, children began each day with two hours of mCIMT, followed by four hours of BT (Figure 1).

During mCIMT, children wore a mitt over their less-affected hand, which restricted use of that hand. Children engaged in intensive therapy to improve the active range of motion, strength, motor control, and sensory awareness of the affected hand. Activities were functional and play-based. Daily structure included: morning gym, fine motor, gross motor, sensory motor, therapeutic feeding, sports, and self-care activities. During training, children performed play-based and functional activities with the affected hand. Example activities included playing card and board games, arts and crafts, and activities that provided sensory stimulation to the affected hand, such as finger painting. Activities also included stretching and strengthening, and reciprocal coordination exercises.

During BT, children did not wear a mitt over the less-affected hand. Children were provided individualized activities that facilitated active use of both hands. Therapists adapted and graded activities and guided children to problem-solve for success. Bimanual activities included self-care (tying shoes, zippering, cutting food), sports activities, and manipulation of classroom tools (cutting with scissors). 

Tailoring of therapy: Activities were selected for each participant based on the child’s preferences, interest, and functional goals. Examples of the children’s preferred interests include sports, arts and crafts, model construction, music, dancing, and computer games. Some examples of functional goals include donning and doffing clothing, using eating utensils, pouring liquid into a cup, carrying a lunch tray, and opening zippered food storage bags. 

### 2.3. Group Allocation

This study began as a clinical program at Blythedale Children’s Hospital, held once annually during the summer. From 2011 to 2015, the Step protocol was used exclusively. In 2016, we decided to add the CB and BC groups. In 2016–2018, only the CB and BC protocols were used. During this time, children were randomized to either of the two groups. Thus, the Step group was not randomized, while the CB and BC groups were randomized. Each cohort was split into two equally sized, age-matched groups. Then, each group was randomized to either the CB or BC interventions. Randomization was done off site by a scientist not otherwise associated with the study.

### 2.4. Outcome Measures

All study participants were evaluated at three time points: before the first day of treatment, within two days of the end of treatment, and two months after treatment. Bimanual performance was tested for in the CB and BC groups after week three of the intervention, when they switched between mCIMT and BT. Three outcome measures were chosen to quantify bimanual performance, motor function of the impaired upper extremity, and functional goal performance. 

The Assisting Hand Assessment (AHA) was used to measure how children use the two hands together. The AHA quantifies how well children with unimanual upper limb impairments use their impaired hand when performing bimanual activities. The AHA shows excellent validity, reliability (0.97–0.99) and responsiveness to change [19]. The test was videotaped and scored by a trained evaluator. Scores were computed as logit-based AHA units. The functionally meaningful difference in score for the AHA is 4 points [20].

The Canadian Occupational Performance Measure (COPM) was used to measure performance and satisfaction levels in functional goals in self-care, productivity, and leisure domains [21]. The COPM is a standardized test in which a child’s caregiver identifies a child’s functional goals during a structured interview [22]. The caregiver rates satisfaction and performance on a scale of 1 (poor) to 10 (excellent), on a maximum of five goals. The minimal clinically important difference is 2 points. Mean performance and caregiver satisfaction scores were analyzed.

The Pediatric Evaluation of Disability Inventory (PEDI) was used to assess each child’s functional independence with activities of daily living. For this study, only the PEDI self-care domain was used, as evaluated by a caregiver. Scaled performance scores ranging from 0 (poor) to 100 (excellent) were used to assess change over time. The MCID is 11 points [23]. It has very good inter-rater reliability, with an intra-class coefficient of 0.7–0.98 [24].

The AHA was conducted by staff who were not therapists in the intervention. The AHA was scored by a blinded, trained, certified evaluator who was not involved in any other aspect of this study. The COPM and PEDI were given by one of the therapists, who may or may not have worked with a particular child.

### 2.5. Statistical Analyses

A group × time repeated-measured analysis of variance (ANOVA) was used to evaluate differences in outcome measures after the intervention and a two-month follow-up, for each intervention group, using SPSS Software (IBM, version 21). Missing data were interpolated based on average changes in measures from pre- and immediate post-intervention to 2 months follow-up. Two-month AHA follow-up data were missing for two children in the Step group, one child in the BT group, and two children in the CB group. Two-month COPM follow-up data were missing for one child in the Step group, one child in the BC group, and two children in the CB group. Two-month PEDI follow-up data were missing for seven children in the Step group, one child in the BC group, and two children in the CB group. Analyses were done with and without the inclusion of the missing data estimates, and the statistical outcomes were not different between these methods. The findings presented below include the missing data estimates. When main ANOVA effects were found, post-hoc analyses were performed, using Bonferroni corrections to correct for multiple comparisons. The Fisher’s exact test was used to compare baseline categorical variables, sex, and side of lesion, among the groups. A *p*-value of less than 0.05 was considered statistically significant.

## 3. Results

Thirty-five children with unilateral CP participated in a six-week intensive occupational therapy program that combined unimanual and bimanual training. Participant demographics and baseline clinical measures are presented in Table 1. We examined differences among baseline characteristics of the groups. There were no significant differences in sex (Fisher’s Exact, *p* = 1.0), lesion side (Fisher’s Exact, *p* = 0.61), age (F(2,32) = 0.48, *p* = 0.63), COPM-Performance (F(2,32) = 2.4, *p* = 0.11), COPM-Satisfaction (F(2,32) = 1.75, *p* = 0.19), or the PEDI (F(2,32) = 2.6, *p* = 0.087). There was a difference in baseline AHA among groups (F(2,31) = 3.7, *p* = 0.037)), which was a limitation of this study. The AHA for the CB group was significantly higher than the AHA for the Step group (*p* = 0.033). The BC group AHA scores were not significantly different from the other groups (*p* > 0.45).

### 3.1. Improvements in Bimanual Hand Function after Intervention

Bimanual hand function was measured before, after, and two months after intervention with the Assisting Hand Assessment (AHA; Figure 2A). There was no main effect of the intervention on AHA scores (F(2,39) = 1.03, *p* = 0.37). We assessed how many children per group met the functionally meaningful difference for the AHA, which is 4 points. In the Step group, 71.4% children improved by 4 or more points, while 70% of children in the BC group and 36% of children in the CB group reached the functionally meaningful difference.

### 3.2. Improvements in Self-Care Skills Independence after Intervention

Changes in self-care skills performance were measured before, after, and two months after intervention with the Pediatric Evaluation of Disability Inventory (PEDI). Caregiver reports of self-care skills performance were obtained. There was a significant improvement in skill performance (Figure 2B) outcomes after intervention (F(2,39) = 4.2, *p* < 0.031). For COPM Performance, the CB group improved slightly more than the BC group (*p* = 0.034), while the Step group did not differ from the CB (*p* = 0.89) or BC (*p* = 0.24) groups. For COPM Satisfaction, there were no significant differences in improvement among the groups (*p* > 0.1).

### 3.3. Improvements in Functional Goal Performance and Satisfaction after Intervention

Changes in functional goal performance and satisfaction were measured before, after, and two months after intervention with the Canadian Occupational Performance Measure (COPM). Caregiver reports of goal performance and satisfaction with performance were obtained. There was a significant improvement in both performance (Figure 2C) and caregiver satisfaction (Figure 2D) outcomes after intervention (Performance: F(2,39) = 19.1, *p* < 0.0001; Satisfaction: F(2,39) = 35.2, *p* < 0.0001) that was retained two months after intervention. These represent clinically meaningful improvements in both functional goal performance and satisfaction.

### 3.4. Midpoint Analysis of Bimanual Function in BC and CB Groups

In the BC and CB groups, we measured bimanual function using the AHA after three weeks of the intervention. We examined whether bimanual function changed differently if children received mCIMT or BT in the first block of the intervention (Figure 3). The AHA significantly improved for both groups after three weeks (F(3,57) = 3.49, *p* = 0.034). There were no significant differences between groups (F(3,57) = 1.31, *p* = 0.28), meaning that the order of mCIMT or BT delivery did not impact efficacy.

## 4. Discussion

This study compared the efficacy of different combinations of mCIMT and BT for improving hand function in children with uCP. The intervention improved self-care skills independence and performance and satisfaction in caregiver-rated functional goals. Improvements were maintained two months after therapy.

While many clinical trials have shown that CIMT and bimanual therapy have equivalent efficacy in children with hemiplegia [6,7,8,9,10], a recent focus has been to select the hypothesized key ingredients from each therapy and combine these two approaches. CIMT provides focused training of the impaired hand, which may be optimal for improving strength of that hand [13,14]. Bimanual therapy has been shown to be slightly better than CIMT at improving functional, bimanual hand use [25]. 

In our study, we did not find clinically meaningful differences in outcomes among the three groups. We had hypothesized that children receiving mCIMT before BT would have the greatest improvements. In our comparison of pre, post, and two-month follow-up time points for all three groups, we found some group differences, but these did not differ by a clinically meaningful amount. Children in the CB group improved in COPM Performance more than the BC group, by a difference on 0.8 points, whereas the clinically meaningful difference is 2 points. Children in the Step group improved in the PEDI more than the BC group, by a difference of 7.3 points, while the clinically meaningful difference is 11 points. Thus, we conclude that the three interventions are not meaningfully different in their efficacy. This could indicate that at the intensity delivered, either type of training was sufficient to drive change. 

Despite the abundance of research into the optimal therapies for children with uCP, all available therapies remain unable to ameliorate impairments. Children, and their families, invest a massive amount of time, effort, and hope in the best available therapies. Nevertheless, these children spend a lifetime with movement impairments. Much more work is needed to develop effective therapies that enable children to sustain long-term improvements in function.

Further study is needed to better understand the optimal combination, schedule, and intensity of therapeutic strategies. There are a wide variety of factors that contribute to movement, and even the best available therapies do not address all factors. For example, many children with uCP have impairments in motor planning and motor imagery [26,27,28,29]. Adding action-observation training to CIMT can improve the efficacy of CIMT [30]. The role of sensory impairments in improvement of movement needs to be further studied, as the sensory system plays an essential role in accurate voluntary movement [31].

Moreover, a better understanding of individual differences in responsiveness to therapy is needed. There is a high variability in responses among children in the studies cited in this manuscript. Efficacy can depend on how engaged a child is in the intervention [32], the intensity of training [33], and a myriad of other factors. One study suggests, for example, that a child may be more responsive to CIMT or bimanual therapy based on whether their impaired hand is controlled via contralateral connections from the injured motor cortex, or via ipsilateral connections from the uninjured motor cortex [34], while others did not find a difference in efficacy based on motor system connectivity [35]. Gender may also affect efficacy, though the key ingredient may be the active engagement of each child in the training. When conducting group interventions, it is important to have a variety of fun activities that will be appealing to children of varied genders, ages, ability levels, and interests. 

There were several limitations to this study that pertain to how the study was done, and how the findings can be interpreted. First, we will discuss limitations regarding how the study was done. The study is underpowered to be an efficacy study or a non-inferiority study. However, our results are consistent with prior studies comparing unimanual and bimanual training [6,9,10,36,37]. A limitation is that the Step group was completed before the CB and BC groups were developed. Ideally, all three interventions would have been tested at the same time, and children would have been randomized to each of the groups. The length of time between the delivery of the three intervention types may have affected the outcomes. Another limitation is the large number of children in the Step group who did not complete the two-month follow-up PEDI evaluation. The PEDI and COPM findings also have a limitation, since therapists associated with the intervention conducted these surveys. Finally, a limitation is that the CB group had a higher baseline AHA score than the Step group.

There are also limitations regarding how our findings can be interpreted. The children in this study were school aged between 4 and 12 years old. We chose this age group because they are capable of remaining engaged in training tasks for 6 h/day, but it is possible that younger children may show greater improvements in a similar intervention, since neuroplasticity is greater in young children [38]. Our findings cannot be generalized to younger children, teens, or adults with uCP.

Moreover, our conclusions are limited by the length of follow-up of this study. We included a two-month follow-up evaluation, but ideally the effects would be sustained long-term. In the future, longer follow-ups, such as six months later, would provide important information about the longevity of improvements. It is possible that our three schedules of the intervention led to different rates of long-term retention of improved function.

In this study, we hypothesized that children who received unimanual training before bimanual training would improve more than children who did not receive unimanual training first. Our findings indicate, however, that the order of training strategies does not significantly affect outcomes. The optimal schedule of training approaches is likely to be specific to an individual’s impairments and other unknown factors. More work is needed to better understand how to optimize a child’s improvements.

## 5. Conclusions

This study compared the efficacy of blocks of mCIMT and BT or progressive shift from mCIMT to BT for improving hand function in children ages 4–12 with uCP. All groups improved equally in self-care skills independence and performance and satisfaction in caregiver-rated functional goals. This interpretation is limited by a small sample size, lack of long-term follow up, and differences in baseline bimanual function among the three groups.

## Figures and Tables

**Figure 1 behavsci-13-00490-f001:**
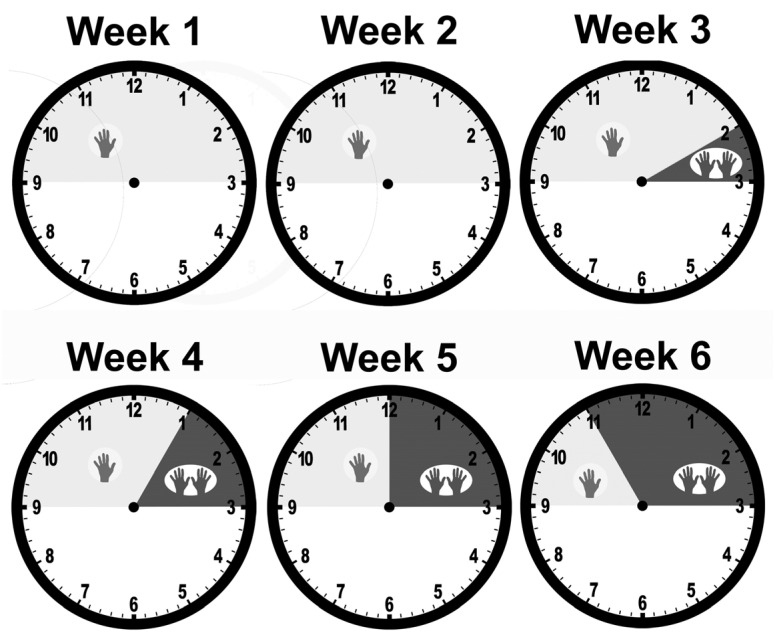
Delivery schedule for the Step group. Each clock represents a day’s schedule, with the same schedule used each day that week. The intervention ran from 9 am to 3 pm, five days per week. During the first two weeks, children received mCIMT for the entire six hours, represented by a single hand icon and a shaded region on the clock, corresponding to the time of day. In week 3, children received mCIMT for 5 h, 9 am to 2 pm, followed by one hour of bimanual training, represented by two hand icons. In each subsequent week of intervention, the duration of mCIMT was reduced by one hour per day, while the duration of BT was increased by one hour per day. Throughout the intervention, mCIMT was always given first in the day, followed by BT. By week 6, children received two hours per day of mCIMT (9 am to 11 am) followed by BT (11 am to 3 pm).

**Figure 2 behavsci-13-00490-f002:**
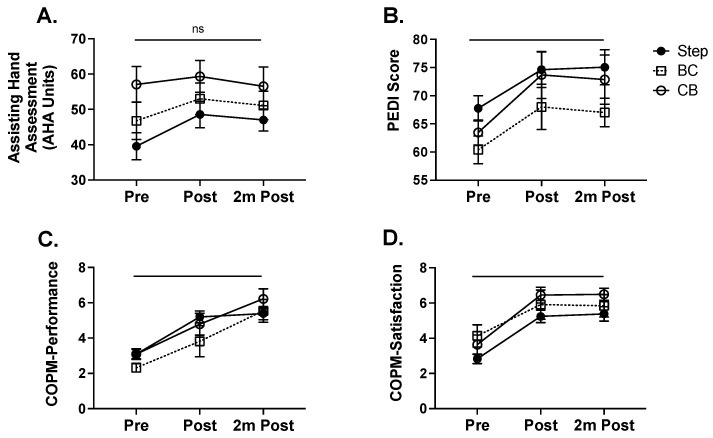
Outcome measures assessed before, immediately after, and two months after the intervention. (**A**). There were no statistically significant differences in the AHA between the time points or groups. (**B**). There was an overall improvement in the PEDI (*p* < 0.031), with the Step group improving more than the BC group (*p* = 0.0022). There was an overall improvement in the COPM Performance (*p* < 0.0001, (**C**)) and in the COPM Satisfaction (*p* < 0.0001, (**D**)). For COPM Performance, the CB group improved more than the BC group (*p* = 0.034).

**Figure 3 behavsci-13-00490-f003:**
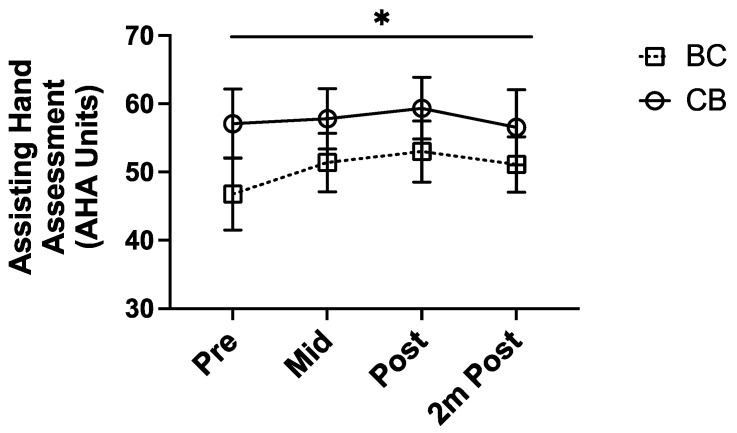
Comparison of bimanual function for the BC and CB groups, including a midpoint measure done at the end of week 3 of the intervention. There was an overall improvement in AHA scores across all time points (*p* = 0.034), with no difference in AHA scores between the groups (*p* = 0.28). * *p* < 0.05.

**Table 1 behavsci-13-00490-t001:** Baseline Participant Characteristics.

Child/Group	Sex	Age (Y, M)	Paretic Side	AHA Baseline	COPM Baseline	PEDI Baseline
Step01	M	3, 11	R	21	3.2 (P), 2.8 (S)	58
Step02	F	7, 9	R	47	3.8 (P), 4.5 (S)	74.7
Step03	M	9, 1	R	53	3.5 (P), 2.5 (S)	72.6
Step04	F	6, 6	R	38	4.2 (P), 3.6 (S)	57.4
Step05	M	8, 1	R	35	3.4 (P), 2.6 (S)	66
Step06	F	12, 5	L	30	4.0 (P), 3.4 (S)	79
Step07	M	8, 1	R	17	4.8 (P), 4.0 (S)	66
Step08	F	10, 4	R	58	3.4 (P), 3.8 (S)	77.3
Step09	F	10, 7	R	44	1.6 (P), 1.8 (S)	74.7
Step10	M	5, 2	L	44	1.8 (P), 2.6 (S)	55.6
Step11	M	6, 7	L	24	2.0 (P), 1.6 (S)	71.7
Step12	M	5, 8	R	27	3.0 (P), 3.4 (S)	61.2
Step13	M	6, 3	R	58	3.0 (P), 2.4 (S)	58.6
Step14	M	9, 5	L	58	1.6 (P), 1.0 (S)	75.9
BC01	M	12, 0	L	50	4 (P), 7.2 (S)	65.3
BC02	F	5, 8	R	59	2 (P), 5 (S)	53.7
BC03	M	5, 3	R	30	2 (P), 4.8 (S)	55.6
BC04	M	4, 9	R	61	1.8 (P), 5 (S)	59.9
BC05	M	5, 2	L	7	1.4 (P), 2.8 (S)	51.7
BC06	F	9, 0	L	52	2.2 (P), 5.8 (S)	61.2
BC07	F	8, 8	R	59	3.2 (P), 3 (S)	75.9
BC08	M	4, 11	L	55	2.4 (P), 1.4 (S)	55.6
BC09	M	5, 9	L	52	1.8 (P), 2.2 (S)	70
BC10	M	5, 5	R	43	2.4 (P), 2.6 (S)	54.9
CB01	M	4, 8	R	87	4.2 (P), 6.8 (S)	63.9
CB02	M	5, 9	R	33	3.4 (P), 6.6 (S)	59.9
CB03	M	10, 2	R	50	1.8 (P), 5.2 (S)	75.9
CB04	F	4, 9	L	65	2 (P), 5.2 (S)	60.5
CB05	F	11, 2	R	50	3.8 (P), 3.4 (S)	60.5
CB06	F	5, 8	R	52	4.6 (P), 4.2 (S)	62.5
CB07	M	11, 0	L	47	2.2 (P), 1.6 (S)	77.3
CB08	M	9, 8	L	52	3 (P), 1.8 (S)	65.3
CB09	F	5, 1	L	66	2 (P), 2 (S)	63.2
CB10	M	7, 6	R	84	4 (P), 2 (S)	56.2
CB11	M	5, 9	R	42	3 (P), 2.2 (S)	53
**Summaries**	**Counts**	**Avg ± SD**	**Counts**	**Avg ± SD**	**Avg ± SD**	**Avg ± SD**
Step (*n* = 14)	9M/5F	7.4 ± 2.3	10R/4L	39.6 ± 14.3	3.1 ±1.0 (P), 2.8 ± 1.0 (S)	67.8 ± 8.3
BC (*n* = 10)	7M/3F	6.8 ± 2.5	5R/5L	46.8 ± 16.7	2.3 ± 0.8 (P), 4.0 ± 1.8 (S)	60.4 ± 7.8
CB (*n* = 11)	7M/4F	7.5 ± 2.7	7R/4L	57.1 ± 16.8	3.1 ± 1.0 (P), 3.7 ±2.0 (S)	63.5 ± 7.4

Abbreviations: Step, stepwise progression group; BC, bimanual training followed by modified constraint-induced movement therapy; CB, modified constraint-induced movement therapy followed by bimanual training; M, male; F, female; R, right; L, left; AHA, Assisting Hand Assessment; COPM, Canadian Occupational Performance Measure; P, performance; S, satisfaction; PEDI, The Pediatric Evaluation of Disability Inventory; Avg, average; SD, standard deviation.

## Data Availability

Requests for data can be made to Kathleen Friel, kaf3001@med.cornell.edu.

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
