# Peer review of "Combining Unimanual and Bimanual Therapies for Children with Hemiparesis: Is There an Optimal Delivery Schedule?"

_behavsci, 2023, doi:10.3390/bs13060490_

Round 1

Reviewer 1 Report

This is an important paper and contributes important new knowledge to the field.  I recommend adding one sentence to the discussion limitation, that the study is neither powered to be an efficacy study nor a non-inferiority study, nevertheless, the findings are consistent with previous studies in these 2 interventions

Author Response

Thank you for your review. We have added this to the discussion of limitations, lines 320-322.

Reviewer 2 Report

Dear authors,

I read with interest your paper dealing with the effect of 3 different delivery schedules of bimanual therapy and constraint induced movement therapy in children with cerebral palsy, two major intensive motor therapies that have already demonstrated their efficacy on upper limb function in these children. The paper is well-written and statistical analyses seems appropriate. However, improvements are needed. There is a lack of clarity in the result part, and the discussion is not developed enough. You need to argue more the interest of this study and what it demonstrated for the implementation of these therapies in clinical practice. I suggest to develop the discussion with key ingredients of intensive motor therapy and literature on bimanual therapies such as HABIT ILE.

1) Introduction: Regarding the rationale of this study, it is not clear if you wanted to demonstrate the interest of combining uni and bimanual training at the same time vs two separate blocks of therapy (group CB and BC), or to demonstrate that the order of the provided therapies is important (mCIMT then BT). Regarding the hypothesis you mentioned, I wonder if the design of the study is appropriate. A cross over analysis would have been sufficient to demonstrate if children who received mCIMT before BT would show a greater improvement in hand function. Please justify your choice or precise your hypothesis.

2) Table 1: Please provide the statistics for comparison between the 3 group to assess if the groups were comparable at baseline (especially AHA scores).

3) Methods: There is redundant information, please simplify (ex: therapy schedule in the 3 groups was explained twice)

4) Who performed the clinical tests scoring (AHA, COPM) of all these children? Is it the same occupational therapists who performed the therapies? Were the assessors blinded to the intervention or the timing of the assessment? It could be a major bias and need to be discussed in the limits of this study.

5) It is written that children were randomized to one of the crossover groups (CB or BC). What about the children included in the group “stepwise incorporation of BT”? A flow chart would have helped to better understand.

6) How do you explain the large number of missing data? Even if missing data were interpolated and you have compared the results with and without interpolation, 7 missing information on 14 children (PEDI scores in group “step”) is a lot to make conclusion on this point. This should be discussed in the limits.

7) P value for significance is missing in the statistical part

8) Results: Regarding AHA results, the authors wrote there is “an overall improvement in AHA scores across all time points”. But, on Figures 2 and 3, it seems that mean AHA score at 2 months was decreased compared to mean AHA score at baseline in the CB group.

9) In the results, the global results on scores improvement in all children are presented but we don’t have the details for group comparison

10) Discussion: This part is too light. The limits are not discussed.

No comments

Author Response

1) Introduction: Regarding the rationale of this study, it is not clear if you wanted to demonstrate the interest of combining uni and bimanual training at the same time vs two separate blocks of therapy (group CB and BC), or to demonstrate that the order of the provided therapies is important (mCIMT then BT). Regarding the hypothesis you mentioned, I wonder if the design of the study is appropriate. A cross over analysis would have been sufficient to demonstrate if children who received mCIMT before BT would show a greater improvement in hand function. Please justify your choice or precise your hypothesis.

Thank you for this suggestion. We have expanded our introduction to include a clearer justification of our study. Added, lines 61-67.

2) Table 1: Please provide the statistics for comparison between the 3 group to assess if the groups were comparable at baseline (especially AHA scores).

            We examined differences among baseline characteristics of the groups. There were no significant differences in sex (Fisher’s Exact, p=1.0), lesion side (Fisher’s Exact, p=0.61), age (F(2,32)=0.48, p=0.63), COPM-Performance (F(2,32)=2.4, p=0.11), COPM-Satisfaction (F(2,32)=1.75, p=0.19), or the PEDI (F(2,32)=2.6, p=0.087). There was a difference in baseline AHA among groups (F(2,31)=3.7, p=0.037)), which is a limitation of this study. The AHA for the CB group was significantly higher than the AHA for the Step group (p=0.033). The BC group AHA scores were not significantly different from the other groups (p>0.45).

This has been added to the results, lines 220-227. The AHA result is also included in the limitations, lines 328-329.

3) Methods: There is redundant information, please simplify (ex: therapy schedule in the 3 groups was explained twice)

We removed the detailed description of the three groups from the introduction, so it is not repetitive when described in the methods.

4) Who performed the clinical tests scoring (AHA, COPM) of all these children? Is it the same occupational therapists who performed the therapies? Were the assessors blinded to the intervention or the timing of the assessment? It could be a major bias and need to be discussed in the limits of this study.

The AHA was conducted by staff who were not therapists in the intervention. The AHA was scored by a blinded, trained, certified evaluator who was not involved in any other aspect of this study. The COPM and PEDI were given by one of the therapists, who may or may not have worked with a particular child. Different therapists work at the same time with all the children. The therapist who did the assessment might not be the same therapist who did the therapy with the children. This indeed is a limitation and is now mentioned as such in the discussion, lines 327-328.

5) It is written that children were randomized to one of the crossover groups (CB or BC). What about the children included in the group “stepwise incorporation of BT”? A flow chart would have helped to better understand.

Thank you for this suggestion. We have added the following section to the methods, lines 161-169.

2.3 Group Allocation

            This study began as a clinical program at Blythedale Children’s Hospital, held once annually during the summer. From 2011-2015, the Step protocol was used exclusively. In 2016, we decided to add the CB and BC groups. In 2016-2018, only the CB and BC protocols were used. During this time, children were randomized to either of the two groups. Thus, the Step group was not randomized, while the CB and BC groups were randomized. During 2016-2018, each cohort was split into two equally sized, age-matched groups. Then, each group was randomized to either the CB or BC interventions. Randomization was done off site by a scientist not otherwise associated with the study.”

6) How do you explain the large number of missing data? Even if missing data were interpolated and you have compared the results with and without interpolation, 7 missing information on 14 children (PEDI scores in group “step”) is a lot to make conclusion on this point. This should be discussed in the limits.

The missing data were from the 2-month follow-up visits. These data were missing because parents/caregivers failed to come back for the follow-up assessment due to various reasons, including busy schedules and difficulty with transportation. Some families did not respond to repeated contacts by study staff, and thus we were never able to schedule the 2-month follow-up for those children. This is mentioned as a limitation, lines 325-326.

7) P value for significance is missing in the statistical part

We added this, lines 215-216. A p-value less than 0.05 was considered statistically significant.

8) Results: Regarding AHA results, the authors wrote there is “an overall improvement in AHA scores across all time points”. But, on Figures 2 and 3, it seems that mean AHA score at 2 months was decreased compared to mean AHA score at baseline in the CB group.

We have edited this to say,”There was an overall improvement in AHA scores.” Line 271. This reflects the ANOVA result when comparing the BC and CB groups, including the midpoint measures. Midpoint measures were not taken in the Step group. The AHA ANOVA for CB vs BC shows an overall effect, but no differences between groups. 

9) In the results, the global results on scores improvement in all children are presented but we don’t have the details for group comparison

We have revised the results to include details of group comparisons. For the AHA, since there was no main effect, post-hoc analyses were not done. For the other measures, post-hoc analyses have been included.

10) Discussion: This part is too light. The limits are not discussed.

We have expanded the discussion substantially, including a paragraph about the study limitations.

Reviewer 3 Report

Combining Unimanual and Bimanual Therapies for children are the good and interesting therapy for children hemiparesis, because neurological disorders therapies should be combined. So it is a good and interesting study, but inclusion and exclusion criteria should be specified and presented. Literature should be expanded.

Author Response

Thank you for your review. Our inclusion and exclusion criteria are presented on lines 61-67.

“Inclusion criteria: 1) unilateral brain injury resulting in impairment of one side of the body, 2) ability to move all joints of affected upper extremity, and 3) ability to comply with study protocol. Exclusion criteria: health problems or uncorrected vision that would interfere with study participation.”

We have also expanded our literature review in the introduction and discussion.

Reviewer 4 Report

The authors present a manuscript describing a study that compares constraint-induced movement therapy (CIMIT) and bimanual therapy (BT) and timeframes for implementing these interventions to determine improvement in children. Overall the manuscript can make a contribution to the field of interventions for children with hemiparesis. The following are recommendations to improve the clarity of the manuscript:

Methods:

-It appears that you are examining children younger than age 12. Is there a rationale for only looking at this age group?

-Figure 1 page 4. It is not clear by the images what is done in weeks 1,2 or how the figures for weeks 3-6 describe the interventions.

-page 5, line 50. Can you provide some examples of the child's preferred interests and functional goals?

Discussion:

-What other factors may have influenced the findings? Child's age (younger than 12 years)? Willingness to participate? Sex?  It would help to hear about your views in these areas.

English appears to be satisfactory but a review would be helpful.

Author Response

Methods:

-It appears that you are examining children younger than age 12..is there a rationale for only looking at this age group?

We started with this age group so that the activities that we designed/selected would be more appropriate for all the children within this age range as we consider their interest, cognitive level, general physical capacity, maturity and psychological factors. Children in their teens have more variability in maturity and interests, and may be less agreeable to actively engage in the intervention due to other activities in their lives.

-Figure 1 page 4. It is not clear by the images what is done in weeks 1,2 or how the figures for weeks 3-6 describe the interventions.

We apologize for the confusion. We have expanded our description of the interventions in the methods and figure caption (lines 131-139) to give better clarity to our methods.

-page 5, line 50. Can you provide some examples of the child's preferred interests and functional goals?

Yes, these have been added, lines 155-159. Thank you for the suggestion. Some examples of the children’s preferred interests include sports, arts and crafts, model construction, music, dancing, and computer games. Some examples of functional goals include donning and doffing clothing, using eating utensils, pouring liquid into a cup, carrying a lunch tray, and opening zippered food storage bags.

Discussion:

*-What other factors may have influenced the findings? Child's age ( younger than 12 years)? Willingness to participate? Sex?  It would help to hear about your views in these areas.

 Willing to participate (ability to accept the constraint mitt) definitely would affect the outcome that is not necessarily related to age. We have children at both end range of our age limit that strongly refused the mitt and participation in activities at the beginning of the intervention. Fortunately, observing their peers convinced these children to participate. These children became active participants in the intervention. We did not see an effect of sex on outcomes. A key to a successful intervention is having plentiful supplies of activities that appeal to a variety of personalities. There was a sufficient variety of activities, such that each child had many choices of engaging activities. We have added this to the discussion.